# Is your batch size the problem? Revisiting the Adam-SGD gap in language modeling

## Abstract

Adam is known to perform significantly better than Stochastic Gradient Descent (SGD) in language models, a phenomenon for which a number of explanations have been proposed. In this work, we revisit this "optimizer gap" through a series of comprehensively tuned baseline training runs for language modeling with Transformers. We exhaustively study how momentum, gradient clipping, and batch size affect the gap between SGD and Adam. Our empirical findings show that SGD with momentum can actually perform similarly to Adam in small-batch settings, if tuned correctly. We revisit existing explanations for Adam's advantage, including heavy-tailed class imbalance, directional sharpness, and Hessian heterogeneity, which struggle to directly explain this phenomenon. Towards bridging this gap in our understanding, by analyzing our Transformer training runs and simple quadratic settings inspired by the literature, we provide new insights, driven by stochastic differential equation models, into the role of batch size on the training dynamics.

## 1 Introduction

The Adam optimizer (Kingma & Ba, 2014) is used pervasively in deep learning, especially when training large language models (LMs) (Grattafiori et al., 2024; Liu et al., 2024; Biderman et al., 2023) and vision Transformers (Dosovitskiy et al., 2020; Kumar et al., 2022) at scale. Industrial practice relies on the success of Adam, and thousands of GPU hours every day are spent at large companies using Adam to train their next-generation large language models.

Even in new sophisticated optimization pipelines looking to dethrone Adam, such as Muon (Jordan et al., 2024), most current implementations (Liu et al., 2025; Shah et al., 2025) rely on plain Adam with weight decay (AdamW, Loshchilov & Hutter (2019)) for critical subsets of parameters, such as normalization layers, text embeddings and prediction heads. This new world is still a bit surprising. Until around 2018, Adam was used only occasionally, while stochastic gradient descent (SGD) with momentum was known to lead to neural networks with better accuracy on unseen data (Wilson et al., 2017), relegating Adam to speed runs and quick comparisons (Goyal et al., 2017). Yet, from the start, language modeling with Transformers required Adam. In fact, Transformer LMs have been reportedly untrainable with SGD (Xiong et al., 2020), especially due to the critical parameters listed above.

Over the years, researchers have offered a number of compelling explanations regarding the remarkable performance of Adam compared to SGD in language modeling, attributing it either to the peculiar noisy nature of text data (Zhang et al., 2020b;a) or the heterogeneous structure (Noci et al., 2022; Zhang et al., 2024a) of the Transformer architecture (Vaswani et al., 2017) — comprising semantically and structurally dissimilar layers. While most hypotheses regarding the Adam-SGD gap can help guide our understanding (Ahn et al., 2024), a particularly crucial insight was recently brought to light by Kunstner et al. (2023): the Adam-SGD gap is also observable in full-batch training, and is hence clear that the stochastic and potentially heavy-tailed nature of stochastic gradients may not be the challenge Adam is able to tackle. Inspired by the latter discussion, we take an **orthogonal approach**:

> *Instead of asking why Adam often outperforms SGD, we wonder:*
> *In which Transformer-based language model training setting, if any, does SGD work?*

In other words, while most recent works try to maximize the gap between SGD and Adam in order to explain it more easily, we here try to minimize it – without sacrificing scale or performance. We

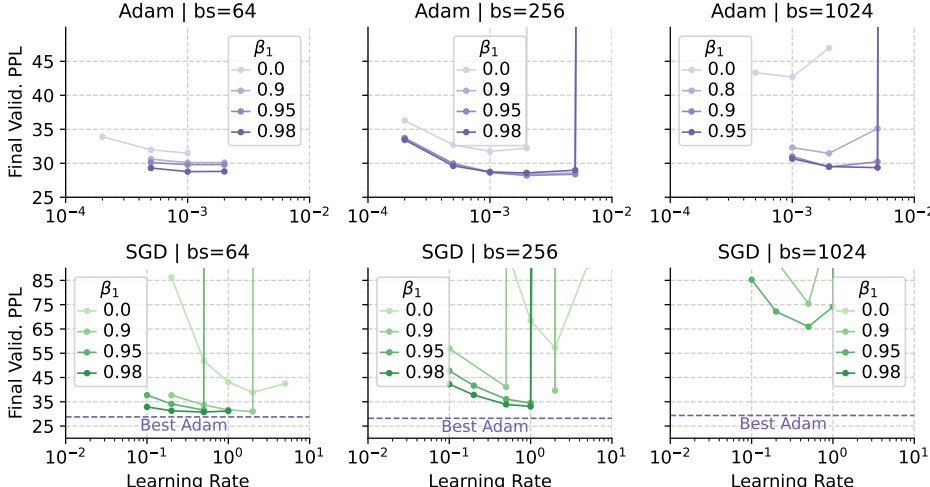

Figure 1: Learning rate and momentum sweep for SGD and Adam across batch sizes under a fixed 1.3B-token compute budget. Experiments use a 160M-parameter model and report perplexities on 100M held-out tokens. **Adam performs consistently across batch sizes,while SGD performs poorly at large batch sizes but improves at smaller ones**. The dashed line marks the best Adam configuration per batch size. Best hyperparameters and perplexities are listed in Table 1.

believe such a view is novel in the literature and can provide valuable insights into the Adam-SGD gap. In particular, it can help identify settings that falsify existing hypotheses about the gap and enumerate necessary criteria that explanations must fulfill. Our contributions are as follows:

- Despite our surprise, we show that LMs can be trained with SGD and achieve performance close to Adam at the same token budget, as long as the batch size is small. We found that this holds even at 1B parameters. While this setting is clearly inconvenient for standard multi-device pretraining, it provides a new lens for understanding the Adam–SGD gap. We note that this finding is consistent with both previously observed trends for small-scale models in the large/full-batch case (Kunstner et al., 2023) and with works observing that adaptive preconditioning affects the critical batch size (Zhang et al., 2019). Yet, taken in isolation, our small-batch results urge us to revisit the theoretical underpinnings for the Adam-SGD gap.

- To inspect this phenomenon, we carefully revisit prior explanations — such as heavy-tailed class imbalance (Kunstner et al., 2024), directional sharpness (Pan & Li, 2023), and Hessian heterogeneity (Zhang et al., 2024a) — in our setup. While our experiments confirm that these explanations can shed light and are useful to describe settings where Adam outperforms SGD, we find that no prior work can directly explain why SGD can outperform Adam at low batch sizes, while achieving satisfactory performance. Notably, in stark contrast with works attributing the gap to heavy-tail noise, we observe that increased stochasticity actually reduces the Adam–SGD gap.

- We enhance our intuition by further studying what makes SGD suboptimal and potentially unstable at high batch sizes. To do this, we present ablations on gradient clipping and learning rate grafting (Agarwal et al., 2020), and inspect their effect on performance.

- Inspired by our observed empirical correspondence between Transformers dynamics and the simplified heterogeneous quadratic setup of (Zhang et al., 2024a) at different noise levels, we leverage this setup to further study why adaptive optimization may have a different batch size sensitivity compared to SGD. Our analysis is rooted in recent works on SDE models (Malladi et al., 2023; Compagnoni et al., 2025b), and our findings and theoretical connections provide evidence of acceleration for Adam in the large batch settings.

Together, our findings paint a new picture of the optimizer gap, and suggest that batch size — and thus the scale and structure of gradient noise — should be explicitly considered in future analyses. Moreover, our results shift the discussion to considerations on the critical batch size of each optimizer, and can provide practical hints in low-resource and small-scale settings, where small batches are common and optimizer memory usage is critical. This last point was recently developed in a concurrent paper by Marek et al. (2025a), where the authors study hyperparameter robustness for small-batch size training in Adam and SGD. In this work, our focus lies primarily in revisiting explanations for the optimizer gap in light of our empirical results.

## 1.1 RELATED WORK

**Class imbalance.** Kunstner et al. (2024) explains the advantage of Adam over SGD on language tasks through the heavy-tailed class imbalance in token distributions. They show that with SGD, loss for rare tokens decrease much more slowly, making training inefficient, while Adam makes steady progress even on low-frequency tokens. Their empirical findings hold across architectures and settings, including non-Transformer architectures and non-textual imbalanced data. This suggests that the performance gap is primarily driven by class imbalance.

**Transformer architecture.** Another line of work focuses on the specific characteristics of Transformer architectures. Zhang et al. (2024a) provide a Hessian-based perspective, showing Transformers have block-heterogeneous Hessian spectrum. In such settings, Adam strongly outperforms SGD, unlike in architectures with more homogeneous Hessian. This holds across modalities, including ViTs, differing from Kunstner et al. (2024).
In contrast, Tomihari & Sato (2025) focus on gradient heterogeneity, explaining Hessian heterogeneity through gradient–Hessian correlations. They show that large disparities in gradient norms across parameters cause challenges for SGD, which Adam's adaptivity addresses. Finally, Zhang et al. (2024b) show that full Adam-style adaptivity is not necessary and can be applied blockwise, as also noted by Zhao et al. (2024), who emphasize the importance of adaptivity for normalization layers.

**Heavy-tailed gradient noise.** Earlier work by Zhang et al. (2020b) asks whether the nature of stochastic gradient noise explains why the Adam–SGD gap exists in Transformers but not in other architectures. They show that Transformers produce gradient noise with heavy-tailed distributions, unlike the nearly Gaussian noise in CNNs, and argue that this degrades SGD performance while Adam remains robust. However, Kunstner et al. (2023) show that noise alone is not the primary cause of Adam's superiority, since the gap persists even in full-batch training and Adam's advantage grows as stochastic noise vanishes. Although their analysis focuses on smaller-scale setups and very large batch sizes, our findings align with extrapolations of their trends to the small-batch regime.

**Optimization trajectories.** Several studies investigate how Adam differs from SGD by analyzing optimization trajectories. Jiang et al. (2022) examine local geometry and define a statistic measuring the uniformity of the Hessian diagonal. On LMs, they find Adam consistently moves through regions with smaller values than SGD with momentum. Rather than examining the entire Hessian, Pan & Li (2023) look at the sharpness along the update direction at each step, showing that Adam makes updates in directions with much smaller sharpness than SGD.

**Evidence from simplified settings.** Recent work shows that the Adam-SGD gap persists even in simplified Transformer architectures. Ahn et al. (2024) demonstrate that the characteristic optimization challenges mentioned above also appear in shallow linear Transformers, models without nonlinear activations, on a linear regression task.

## 2 ADAM VS. SGD: EFFECTS OF HYPERPARAMETERS AND TRAINING REGIMES

To systematically investigate the performance gap between Adam and SGD, we conduct a series of experiments in language modeling using a conventional Transformer architecture. Our goal is to understand how this gap evolves under various training regimes and hyperparameter configurations.

### 2.1 EXPERIMENTAL SETUP

We conduct most experiments on the SlimPajama (Soboleva et al., 2023) dataset using a 160M-parameter nanoGPT (Karpathy, 2022) model, enhanced with recent improvements. Full model details are provided in Appendix A. We also experiment with larger models, up to 1B parameters in a Pythia configuration (Biderman et al., 2023), and on the Fineweb dataset (Penedo et al., 2024).

All experiments are conducted without weight decay. Global gradient norm clipping is applied to raw gradients for both SGD (with momentum) and Adam. Other experiment-specific details are described in the following subsections, with additional information available in Appendix A.

### 2.2 EFFECT OF BATCH SIZE ON THE ADAM-SGD GAP

We first study how the gap between Adam and SGD changes with batch size under a fixed compute budget, when momentum and learning rate are tuned.

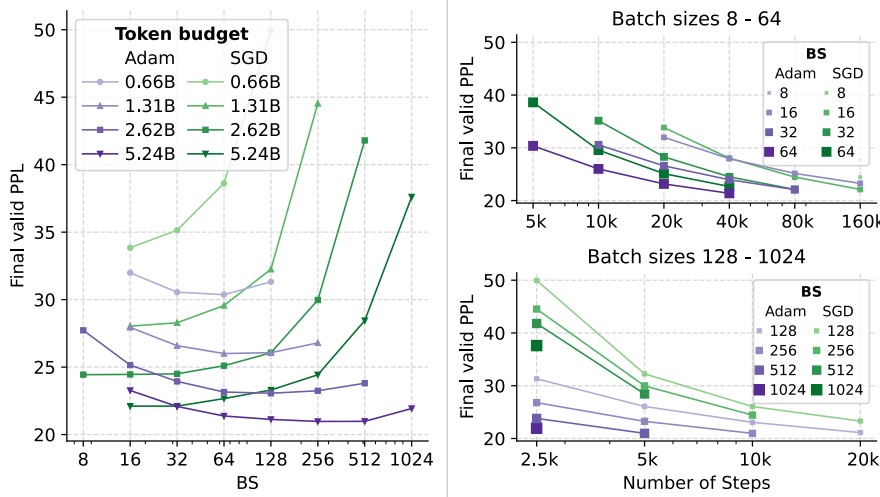

Figure 2: SGD (green) and Adam (purple) performance across batch sizes. Left: fixed token budget (darker colors – more tokens); the gap increases with batch size across all token budgets. Right: fixed number of steps (darker colors – larger batch sizes); the gap decreases with the number of steps. SGD improves with longer training and can match Adam, given a sufficiently small batch size.

**Setup.** All experiments use a sequence length of 512, a fixed token budget of 1.3B tokens, and a cosine learning rate scheduler (Loshchilov & Hutter, 2016) with 10% warmup. We compare three batch sizes: 64, 256, and 1024. The learning rate and momentum values are tuned for both optimizers at a batch size of 256. A sweep is performed over 5 learning rates and momentum values of 0.9, 0.95, and 0.98, including runs without momentum. High momentum values are motivated by findings from Zhao et al. (2024), where SGD performs best with momentum 0.98. For Adam, we fix $\beta_2 = 0.95$. Based on the optimal learning rate found at batch size 256, we scale down the learning rate grid for batch size 64 and scale it up for batch size 1024, sweeping over 3 values in each case. Results are reported in Figure 1. Some settings become unstable at very large learning rates, where one run may succeed, even if the median run diverges. In those settings, we report runs at the largest stable learning rate as optimal.

**Results.** Adam shows similar performance across batch sizes under a fixed token budget, as shown in Figure 1. Surprisingly, *SGD achieves performance close to Adam with small batches*, but the gap grows as batch size increases. For both SGD and Adam, momentum becomes crucial once batch size increases, as noted by Kunstner et al. (2023) and Zhao et al. (2024).

**Additional observations.** We find that using a relatively small sequence length of 512 is not a crucial factor in these dynamics. As we show in the next section, qualitatively similar behavior occurs at a sequence length of 2048, as long as the number of tokens per iteration is held constant. This suggests that performance differences can be attributed to the *effective batch size (in tokens)* rather than sequence length alone. Additionally, we observe that gradient clipping acts differently across batch sizes and is more important at larger batches (see Appendix B.1). Finally, we observe that warm-up length is not a confounder – sweeping 5–20% warm-up schedules in our cosine with warmup scheduler does not affect these dynamics.

## 2.3 ARE LARGE BATCH SIZES THE PROBLEM, OR IS IT THE NUMBER OF STEPS?

Our previous experiments show that SGD in small-batch settings can achieve performance close to Adam. Crucially, note that in Figure 1 all methods see a total of 1.3B tokens. This implies that, e.g., at batch size 1024, methods perform 1/16 of the steps compared to batch size 64. This observation raises an important question: does SGD truly break at large batch sizes, or is it simply slower to converge, compared to Adam, at higher batch sizes? In other words, *can SGD reach Adam-level performance even at higher batch sizes, if given more training steps?*

To investigate this, we compare performance across batch sizes under two training regimes: **(1) fixed token budget** and **(2) fixed number of steps**. This comparison allows us to separate the impact of slow SGD convergence from the inherent difficulty of optimizing in large-batch regimes.

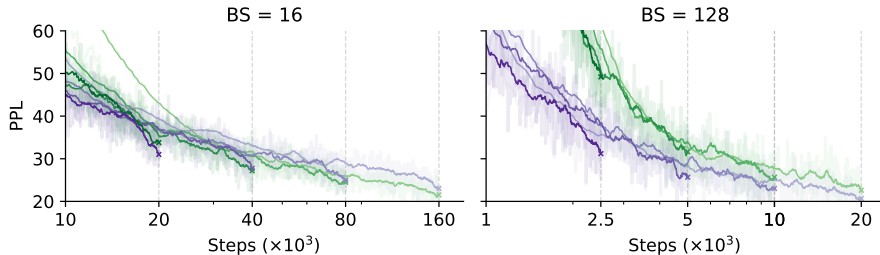

Figure 3: Perplexity during training for SGD (green) and Adam (purple) across training lengths in small- and large-batch settings. High-opacity lines show EMA; low-opacity lines show raw values. The gap shrinks with training, and for small batches, SGD outperforms Adam at the longest run.

**Setup.** The experimental setting is as in the previous section, except we increased the sequence length to 2048 to study its effect. We train models across batch sizes 8–1024 and steps 2.5k–160k. Total token budgets range from approximately 650M to 5.2B: models using larger batches are not trained for the largest number of steps, while models using smaller batches are trained for more steps. We switch from the cosine scheduler used previously to a WSD scheduler (Hägele et al., 2024) to better compare runs before learning rate decay begins, using a fixed 2000-step warmup. For SGD, we use momentum $\beta = 0.98$; for Adam, we set $\beta_1 = \beta_2 = 0.95$. See Appendix A.2 for full details.

**Results.** The left panel of Figure 2 clearly shows that, at a fixed token budget, Adam improves with larger batch sizes, up to some **critical batch size**. In contrast, SGD shows a drastically opposite trend — performance consistently degrades as batch size increases. Under a fixed token budget, matching performance between Adam and SGD is conditional on using very small batch sizes, leading to significantly longer training and poor memory usage. This result highlights a key limitation of SGD: it is highly inefficient in realistic model training, where large batches are required for efficiency.

In the right panel of Figure 2, we show performance after training with various numbers of steps. The gap between Adam and SGD grows with batch size, but SGD improves significantly with more steps and can eventually match or even outperform Adam with long enough training. To illustrate this, we report perplexity during training for SGD and Adam with batch sizes 16 and 128 in Figure 3, and show the same plots for other batch sizes in Appendix A.2.

This observation shows that SGD is not necessarily unable to optimize in large-batch settings, just very slow to converge. Or, to put it differently: while Adam can accelerate (in terms of progress per step) with increased batch size, SGD cannot — its **critical batch size** is close to 1.

**Scaling experiments.** To test whether our findings persist at scale and across datasets, we experiment with larger models and the FineWeb dataset (Penedo et al., 2024) in addition to SlimPajama. We repeat the same experiments — varying token budgets and number of training steps — first by changing the dataset for the 160M model to FineWeb, and then by scaling the model to 250M on FineWeb. Setup details results are reported in Appendix A and Appendix B.2, showing that our core claims hold when scaling up the model and switching datasets.

To further test whether SGD can outperform Adam at larger scales, we scale the model to 410M and 1B. For both, we only tune the learning rate; SGD momentum is set to $\beta = 0.98$ and Adam $\beta_1 = \beta_2 = 0.95$. We use the largest batch size that fits on a single NVIDIA A100 80GB GPU without gradient accumulation. This results in batch size 8 for the 410M model (SlimPajama, sequence length 2048) and batch size 16 for the 1B model (FineWeb, sequence length 1024). Full training details are in Appendix A. Trajectories for both models are shown in Figure 4, demonstrating that **SGD can outperform Adam even at 410M and 1B parameters**.

## 2.4 TUNING ADAM IN SMALL-BATCH SETTINGS

In previous sections, we showed that SGD can perform on par with Adam or even outperform it. One important detail is that even in Section 2.2, where hyperparameters are carefully tuned, we did not explore the effect of tuning Adam's $\beta_2$. In all experiments so far, we used default values recommended in the literature. We believe working in this setup is valuable, since $\beta_2$ is often kept constant while scaling (Biderman et al., 2023; Wortsman et al., 2023; Zhang et al., 2025), except in critically large scenarios (Molybog et al., 2023). However, Zhang et al. (2022) shows that higher $\beta_2$ values substantially improve small-batch training, and Marek et al. (2025b) highlights the importance

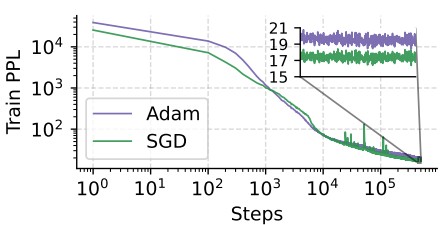 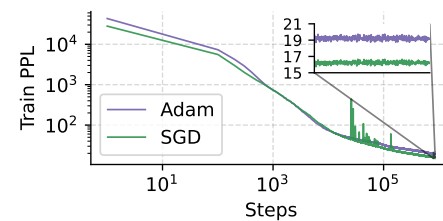

(a) 410M model on SlimPajama – 1.5 days of training.     (b) 1B model on FineWeb – 5 days of training.

Figure 4: SGD can outperform Adam even at 410M and 1B scales in small-batch regimes.

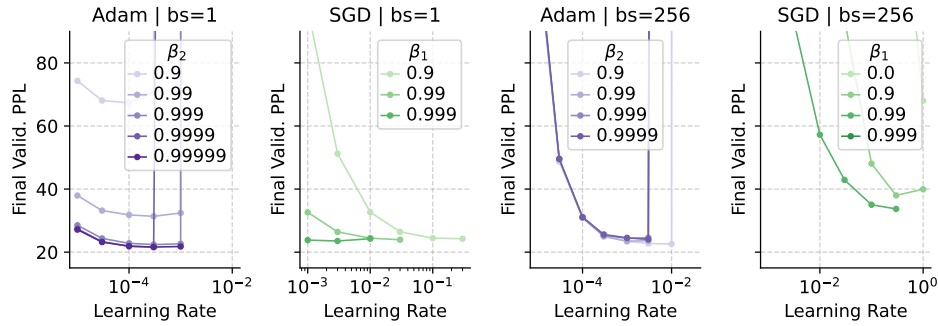

Figure 5: Effect of Adam's $\beta_2$ on the gap for batch sizes 1 and 256. High $\beta_2$ improves Adam's small-batch performance, while large-batch performance is insensitive.

of scaling $\beta_2$ in this regime. In this section, we study this effect. The experimental setup is similar to Section 2.3, and we compare two extreme cases, batch size 1 and 256, under the same token budget. We fix Adam's $\beta_1 = 0.9$, vary $\beta_2$, and tune the learning rate for each. To analyze the gap, we also tune momentum $\beta_1$ for SGD. Details are in Appendix A.3. From Figure 5, for batch size 1, we observe that Adam requires higher $\beta_2$ values, larger than 0.99, to achieve good final perplexity. In contrast, for batch size 256, Adam's final performance is not sensitive to $\beta_2$. Compared to SGD, Adam consistently achieves lower final perplexity, even at batch size 1. Although SGD does not outperform Adam, the gap shrinks substantially as batch size decreases, showing that SGD remains competitive in the small-batch regime, even with carefully tuned Adam $\beta_2$. Results for batch size 8, along with full perplexity curves during training, are provided in Appendix B.3.

## 3 REVISITING PRIOR EXPLANATIONS

Several recent works have proposed explanations for Adam's advantage over SGD through the lens of data or architectural properties (see Section 1.1). All these explanations improve our understanding of the performance gap, yet most are limited to scenarios where the gap between Adam and SGD is pronounced. Our goal is to revisit these explanations through the lens of our findings in Section 2 and ask: *Can they also account for strong SGD performance in small-batch settings?*

In this section, we focus on two approaches: the heavy-tailed class imbalance hypothesis (Kunstner et al., 2024) and the explanation based on heterogeneous Hessian structure (Zhang et al., 2024a). We also analyze the explanation proposed by Pan & Li (2023), with detailed discussion in Appendix C, which shows that directional sharpness correlates with the gap but does not explain it or relate to batch size. While no discussion in the literature fully accounts for our empirical evidence, the heterogeneous toy quadratic example of Zhang et al. (2024a) offers particularly valuable insights, which we develop in Section 4.1.

### 3.1 HEAVY-TAILED CLASS IMBALANCE

Prior work by Kunstner et al. (2024) attributes Adam's advantage over SGD to heavy-tailed class imbalance in token distributions, showing that SGD has difficulty optimizing rare (least common) tokens. We follow their methodology and group all tokens from the training set into 10 frequency

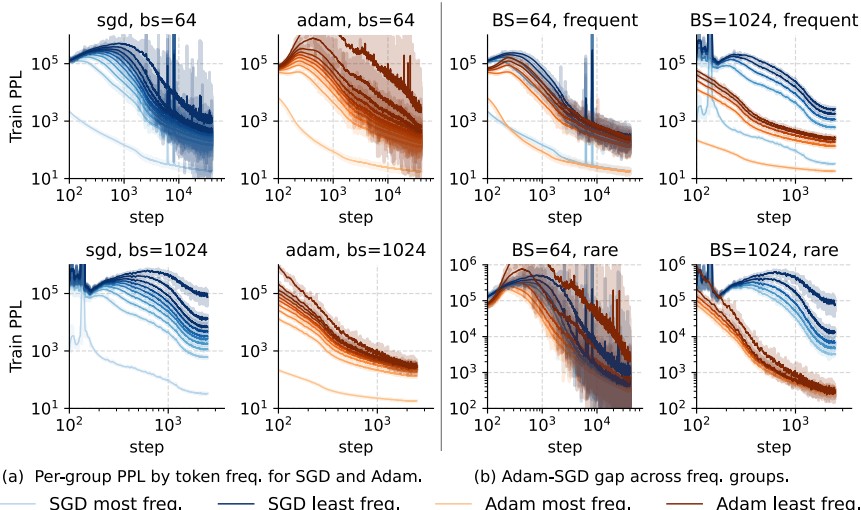

(a) Per-group PPL by token freq. for SGD and Adam.  (b) Adam–SGD gap across freq. groups.

— SGD most freq.  — SGD least freq.  — Adam most freq.  — Adam least freq.

Figure 7: **(left)** Perplexity during training for Adam and SGD in small- and large-batch settings, computed per frequency group. SGD shows a larger gap in the large-batch setting, while the opposite holds for Adam. **(right)** Comparison of the Adam–SGD gap across frequency groups, with frequent tokens on the top row and rare ones on the bottom. SGD underperforms Adam in large batches, especially on rare tokens. This effect is not present in the small-batch setting.

groups, from the first group, which contains the 10% least frequent tokens, to the last group, which contains the 10% most frequent ones.

We apply this analysis to the setting from Section 2.2, comparing batch sizes 64 and 1024, where SGD performs drastically differently, using runs with the optimal combination of $\beta_1$ and learning rate for each case. We find that class imbalance exists in both cases: the persistence of low- and high-frequency tokens is similar, as shown in Figure 6. However, this does not appear to cause problems for small-batch SGD, suggesting that *class imbalance alone does not imply an Adam-SGD gap across all training regimes*.

We further compute perplexity separately for each frequency group and report it over training. From Figure 7 (a), we observe that both optimizers make faster progress on more frequent tokens in all settings, as expected. The relative difference in perplexity between frequency groups is more significant for SGD in the large-batch setting than for the small, while the opposite holds for Adam.

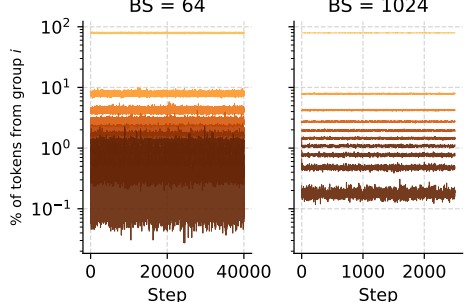

Figure 6: Batch token distribution for batch sizes 64 and 1024. Lighter colors show less frequent tokens. Statistics at lower batch sizes are noisier but of similar magnitude.

Comparing Adam and SGD across frequency groups in Figure 6, we observe that in the large-batch setting, SGD underperforms Adam across all groups, as shown in Figure 7 (b). However, the gap is notably more significant for less frequent tokens, which aligns well with findings from Kunstner et al. (2024), suggesting that rare tokens could be more challenging for SGD in imbalanced settings. In contrast, this effect is not observed in the small-batch regime. Here, not only is the overall Adam-SGD gap small (as seen in Section 2), but the gap across token frequencies is also small. We would expect this problem with SGD to hold independent of batch size, but in settings where SGD works well, the issue disappears.

## 3.2 HESSIAN HETEROGENEITY

From the line of work focusing on the architectural properties of Transformers, Zhang et al. (2024a) argue that the block-wise heterogeneity of the Hessian spectrum is a key factor behind Adam's strong performance and the weakness of SGD. They propose that, based on the Hessian structure at initialization, it is possible to predict whether SGD will perform well, **offering an explanation**

**that is invariant to batch size**. To further explore the effect of batch size on heterogeneous problems, we revisit the simplified quadratic setting from their work and extend it by including batch size variation. We compare optimization on problems with homogeneous (CNN-like) and heterogeneous (Transformer-like) Hessians, where both share the same eigenvalue spectrum. We train with SGD and Adam using a cosine learning rate schedule and no clipping.

From Figure 8, we observe the following:

- Across batch sizes, the largest Adam–SGD gap occurs in the heterogeneous setting. As noted by Kunstner et al. (2023), a similar pattern appears for signed momentum (Bernstein et al., 2018), which we develop further in Section 4.1.
- Adam benefits from larger batch sizes in both homogeneous and heterogeneous Hessian problems, whereas SGD shows little improvement. Details of the setting are provided in Appendix F.

In summary, higher batch sizes boost performance for both SignSGD and Adam, **regardless of heterogeneity**. While heterogeneity amplifies the Adam–SGD gap, this shows that the phenomenon we study is not limited to heterogeneous settings, highlighting that the landscape structure plays a less crucial role, which we further develop in Section 4.1.

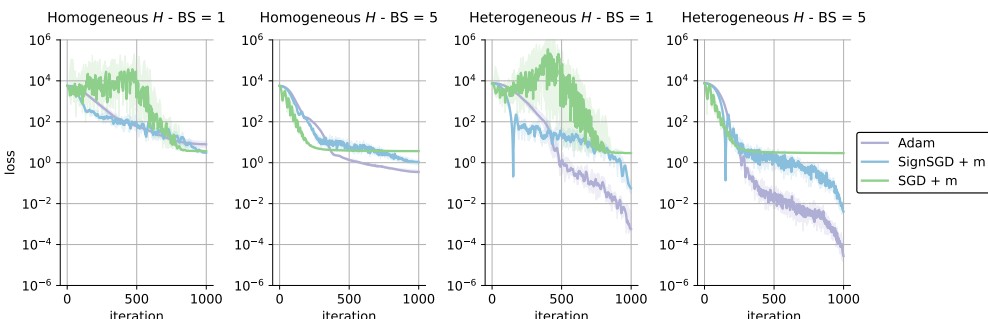

Figure 8: Adam–SGD performance gap across batch sizes in quadratic models from Zhang et al. (2024a). Adam benefits from larger batches in both, more so in Heterogeneous. Learning rates are tuned so to give similar performance at Homogeneous batch size 1. Shown is mean and 2-sigma standard deviation for 10 runs.

## 4 UNDERSTANDING HOW PERFORMANCE RELATES TO BATCH SIZE

In Section 3, we saw that while prior work sheds light on the Adam-SGD gap in the large-batch regime, it remains unclear how batch size itself factors into these explanations. Towards gaining more insights, we proceed as follows:

- We approach this from the SGD angle and ask: *What goes wrong for SGD in large-batch settings that does not appear at small batch sizes?* To investigate, we separate which component of the SGD update is more problematic — its direction or magnitude — focusing on the setting from Section 2.2 with batch size 1024 and the optimal combination of $\beta_1$ and learning rate. Using grafting and adaptive clipping, we find that the main issue is the *update direction* rather than magnitude. Full discussion and experimental results are provided in the Appendix D.
- In Section 4.1 we take a different approach, one based on noise statistics and adaptivity in a setup which is non-specific to the Transformer architecture. This analysis is inspired by the results in Figure 8, showing how adaptive methods may profit from large batch sizes regardless of the Hessian structure. Using theoretical tools, we prove here that while SGD performance in early training is dominated by number of iterations, the dynamics of signed momentum methods (cf. Figure 8) showcase a strong dependency on batch size from the very first iterations.

### 4.1 THEORETICAL INSIGHTS

Towards explaining the phenomena observed in this paper — and specifically the quadratic example in Figure 8 — we provide an analysis based on results around the interaction between noise and

adaptivity in (Compagnoni et al., 2025b). The discussion below is rooted on the observed similarity between behaviors of Adam and SignSGD in Figure 8, as well as recent literature on their relation (Kunstner et al., 2023; Jordan et al., 2024).

Let $X$ denote the model parameters and $\gamma$ denote a batch of size $B$. We denote the stochastic gradient as $\nabla f_\gamma(x) := \frac{1}{B} \sum_{i \in \gamma} \nabla(f_i(x))$ and by $\Sigma(x)$ the noise covariance at batch size 1. The stochastic differential equation (SDE) approximation of SGD reads (Mil'shtein, 1986; Liu et al., 2021)

$$dX_t = -\nabla f(X_t)dt + \sqrt{\frac{\eta \Sigma}{B}}dW_t, \tag{1}$$

We now state a recent result showing that the drift of signed updates – driving performance in early training – has an extra dependency on the batch size. A proof sketch is provided in Appendix E.

**Theorem 1** ((Compagnoni et al., 2025b)). *Under the assumption of i.i.d. Gaussian noise with diagonal covariance (and, with minor modifications, for other noise structures and non-diagonal covariance), the following SDE provides a 1-weak approximation (Mil'shtein, 1986) of SignSGD:*

$$dX_t = -\operatorname{erf}\left(\sqrt{\frac{B}{2}}\Sigma^{-\frac{1}{2}}\nabla f(X_t)\right)dt + \sqrt{\eta}\sqrt{I_d - \operatorname{Diag}\left(\operatorname{erf}\left(\frac{\sqrt{B}\Sigma^{-\frac{1}{2}}\nabla f(X_t)}{\sqrt{2}}\right)\right)^2}dW_t, \tag{2}$$

*where the error function* $\operatorname{erf}(x) := \frac{2}{\sqrt{\pi}} \int_0^x e^{-t^2} dt$ *and the square are applied component-wise.*

While Compagnoni et al. (2025b) provide a similar result for the Heavy-tail noise setting, the Gaussian case already highlights a crucial distinction between signed gradient methods and classical SGD.

**Takeaway.** Recall that $\operatorname{erf}$ is linear on a large interval around zero. The local update of parameters is then driven by $-\operatorname{erf}\left(\sqrt{\frac{B}{2}}\Sigma^{-\frac{1}{2}}\nabla f(X_t)\right)dt$ in the signSGD case, while in the SGD setting, this term is simply $-\nabla f(X_t)dt$. When everything else is kept constant, increasing the batch size $B$, increases the drift in the direction of the negative gradient by $\sqrt{B}$, up until the saturation point of $\operatorname{erf}$, i.e. a critical batch size.

This analysis provides evidence for our results: *using large batch sizes accelerates convergence* (larger drift) in signSGD (and likely also in closely-related algorithms, like Adam), while the performance of SGD in early training is batch-size agnostic and hence driven by the number of iterations (see Figure 9).

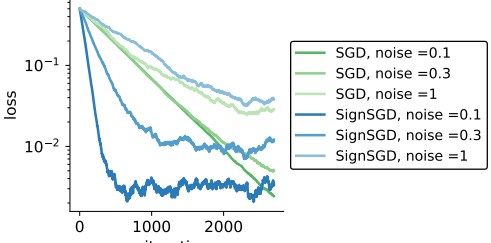

Figure 9: Illustration of theory presented in the section. Optimizing $f(x) = \frac{1}{2}\|x\|^2$, $x \in \mathbb{R}^{100}$. All methods use same learning rate of $1e-3$ and no momentum. Shown is performance under different injected Gaussian noise standard deviation. SGD in **early training is dominated by the drift** component, which is independent of noise – progress is driven by number of iterations. For SignSGD, noise (hence batch size in the more general case) directly affects drift and early progress.

## 5 DISCUSSION

Is it impossible to train language models with SGD? In this work, we show that there exist settings, namely small batch sizes with carefully tuned momentum, where SGD is competitive, even for 1B-scale language models. These findings are interesting in their own right for small-scale training runs, for example, on commodity GPUs, where memory is limited. Yet, these findings also crucially inform the space of possible theories for the optimizer gap between Adam and SGD. We revisit a number of promising theories from the literature based on our findings and find that they have limited explanatory power. We argue that, instead, the effect of batch size is a symptom of the importance of gradient noise for this question, and discuss a stronger explanation based on SDEs.

We believe that future theoretical work might be able to better explain the Adam-SGD gap as a function of the batch size. A promising direction is recent work on $(L_0, L_1)$-smoothness (Zhang et al., 2020a), specifically in the context of signed gradient descent (Compagnoni et al., 2025a), as well as $\ell_\infty$ geometry (Xie & Li, 2024; Xie et al., 2024) of transformer models.

## ACKNOWLEDGMENT OF AI-ASSISTED TOOLS

AI-assisted editing tools were used to check grammar.

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

# Appendix

## A    FURTHER EXPERIMENTS AND EXPERIMENTAL DETAILS

For pre-training Transformers on Causal Language Modeling, we use a setup that builds upon the nanoGPT (Karpathy, 2022) implementation, augmenting it with Rotational Positional Embedding (Su et al., 2024), RMSNorm (Zhang & Sennrich, 2019), and SwiGLU (Shazeer, 2020). We do not adopt QK normalization or $z$-loss, as those modifications are quite recent. All our models have a vocabulary size of 50280 and make use of GPT-Neox tokenizer (Black et al., 2022). We adopt an enhanced training recipe, made popular by large language models such as LLaMa (Touvron et al., 2023). These modifications include: training in bfloat16; employing a linear learning rate warm-up for 10% of the training steps (unless specified otherwise), followed by either cosine annealing to $1e - 5$ of WSD (Hägele et al., 2024). Global norm clipping is used (unless specified or ablated upon) for gradients with norms above 1 (on the raw gradient, as a first step). We have no weight tying between the embedding and the last linear layer. Validation perplexity always refers to a separate subset consisting of 100M tokens.

### A.1    EXPERIMENTAL SETUP

**Computational Resources.**    All experiments use a single NVIDIA A100-SXM4-80GB.

**Code.**    All our runs use the repository `https://github.com/Niccolo-Ajroldi/plainLM`.

**Datasets.**    We test our claims on both the SlimPajama (Soboleva et al., 2023) and Fineweb (Penedo et al., 2024) datasets.

**Model settings (12 Layers, 160M).**    We use the same configuration as (Biderman et al., 2023): `https://github.com/EleutherAI/pythia/blob/main/models/160M/pythia-160m.yml`

- *Layers:* 12 Transformer (Vaswani et al., 2017) layers
- *Attention heads:* 12
- *Hidden size:* 768
- *Attention implementation:* Flashattention (Dao et al., 2022).
- *MLP type:* SwiGLU (Shazeer, 2020) with expansion factor 8/3.
- *Backbone:* PreLN Transformer (Xiong et al., 2020) with skip connections.
- *Normalization:* RMSnorm (Zhang & Sennrich, 2019) for both Attention and MLP.
- *Position embeddings:* Rotary embeddings (RoPE) to 25% of dimensions ((Su et al., 2024))
- *Initialization:*    the MLP and Attention output weights are initialized with variance $0.02/\sqrt{2\#\text{layers}}$ (scaling also similar to (Radford et al., 2019)). All other weights (comprising embeddings) are initialized with a standard deviation of 0.02 (Nguyen & Salazar (2019); Wang & Komatsuzaki (2022), Sec. 2.2). Biases are always initialized at zero.
- *Precision:* Mixed precision FP16 enabled.
- *Dropout:* Disabled for both hidden and attention layers (see also Chowdhery et al. (2023)).

**Model settings (250 M, 24 layers).**    We keep it identical to the setting above, and just increase the number of layers to 24.

**Model settings (410 M).**    We use the same setting as (Biderman et al., 2023), configuration can be found here: `https://github.com/EleutherAI/pythia/blob/main/models/410M/pythia-410m-deduped.yml`

- *Layers:* 24 Transformer layers

- *Attention heads:* 16

- *Hidden size:* 1024

- Other settings as 160M parameters.

**Model settings (1B).** We use the same setting as (Biderman et al., 2023), configuration can be found here: `https://github.com/EleutherAI/pythia/blob/main/models/1B/pythia-1b-deduped.yml`

- *Layers:* 16 Transformer layers

- *Attention heads:* 8

- *Hidden size:* 2048

- Other settings as 160M parameters.

## A.2 HYPERPARAMETER TUNING FOR SECTION 2.3

Combined, the experiments in this section account for full training (at different token budgets) of more than 250 language models at different scales and batch sizes. Every reported result is relative to the best learning rate in our grid, defined for each setup.

**Small-scale experiments.** We consider SGD with $\beta = 0.98$ and global clipping before applying momentum. For Adam, we use the setting $\beta_1 = \beta_2 = 0.95$. Both settings are suggested by the sweep in Figure 1 and recent literature (Zhang et al., 2025; Orvieto & Gower, 2025; Zhao et al., 2024).

- For Figure 2 and Figure 3 (SlimPajama, 160M), we choose a sequence length of 2048. Inspired by the careful tuning of Figure 1, we consider the learning rate grid $[0.25, 0.5, 1.0]$ for SGD and $[0.001, 0.002, 0.004]$ for Adam.

- For Figure 15 (Fineweb, 160M), we choose a sequence length of 1024. Our learning rate grid here is the same as for SlimPajama (previous point). As a sequence length of $160k$, given our lack of experience with extremely low batch sizes (shorter sequence length), we operate on a slightly larger grid: $[0.0001, 0.0003, 0.001, 0.003]$ for Adam and $[0.03, 0.1, 0.3, 1]$ for SGD.

- For Figure 16 (SlimPajama, 250M - 24 layers), we choose a sequence length of 2048 and we also operate on a larger grid: $[0.0001, 0.0003, 0.001, 0.003]$ for Adam and $[0.03, 0.1, 0.3, 1]$ for SGD.

**Medium scale experiments.** For all SGD runs, we use $\beta = 0.98$. For Adam, we use the standard choice (0.9, 0.95) (Biderman et al., 2023). All our runs use global norm clipping and no weight decay.

- **410M model** (Figure 4a): We train with sequence length 2048, for $500k$ steps on SlimPajama. Learning rate grid is $[1.25e-4, 2.5e-4, 5.0e-4, 1.0e-3]$ for Adam and $[0.125, 0.25, 0.5, 1]$ for SGD. The sweep results are presented in Figure 10a.

- **1B model** (Figure 4b): We train with sequence length 1024, for 850k steps on Fineweb. Learning rate sweep, shown in Figure 10b uses $[6.25e-5, 1.25e-4, 2.5e-4, 5.0e-4, 1.0e-3]$ for Adam and $[0.0625, 0.125, 0.25, 0.5, 1]$ for SGD.

## A.3 HYPERPARAMETER TUNING FOR SECTION 2.4

The experimental setup here follows Section A.2. We train a 160M model on SlimPajama, with sequence length 2048 and the WSD scheduler. We consider batch sizes 1, 8, and 256 under a token budget of 3.2B tokens.

For Adam, we fix $\beta_1 = 0.9$, vary $\beta_2 \in [0.9, 0.99, 0.999, 0.9999, 0.99999]$, and tune the learning rate for each configuration. For SGD, we similarly tune the momentum $\beta \in [0.9, 0.99, 0.999]$. The learning rate grids are: for Adam $[1.0e-5, 3.0e-5, 1.0e-4, 3.0e-4, 1.0e-3, 3.0e-3, 1.0e-2, 3.0e-2]$ and $[1.0e-3, 3.0e-3, 1.0e-2, 3.0e-2, 1.0e-1, 3.0e-1, 1.0]$ for SGD.

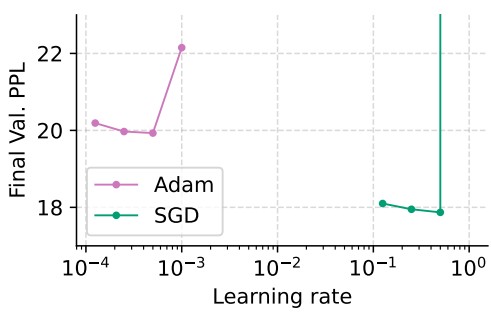 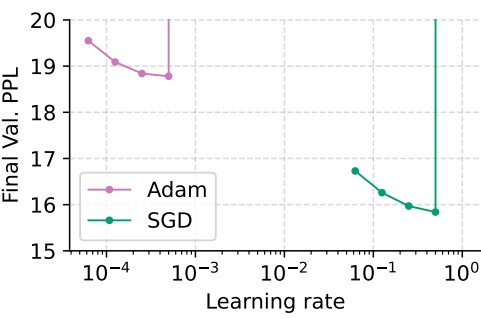

(a) 410M model on SlimPajama (seq. length 2048, batch size 8, 500k steps)

(b) 1B model on FineWeb (seq. length 1024, batch size 16, 850k steps)

Figure 10: Learning rate sweep for 410M and 1B models. Trajectories for the optimal learning rate are shown in Figure 4.

## B  ADDITIONAL RESULTS

In addition to Figure 1, we report the validation perplexity for the best-performing $\beta_1$ and learning rate combination for both Adam and SGD across batch sizes in Table 1. The experimental setting is described in Section 2.2.

Table 1: Best validation perplexities and corresponding hyperparameters for Adam and SGD across batch sizes. Results correspond to the sweep shown in Figure 1.

| Batch Size | Optimizer | PPL | Hyperparameters | |
|---|---|---|---|---|
| 64 | Adam | **28.77** | $\beta_1 = 0.98$, | $\mathrm{lr} = 1\mathrm{e}{-3}$ |
| | SGD | **30.76** | $\beta_1 = 0.98$, | $\mathrm{lr} = 5\mathrm{e}{-1}$ |
| 256 | Adam | **28.20** | $\beta_1 = 0.95$, | $\mathrm{lr} = 2\mathrm{e}{-3}$ |
| | SGD | **33.08** | $\beta_1 = 0.98$, | $\mathrm{lr} = 1\mathrm{e}{+0}$ |
| 1024 | Adam | **29.36** | $\beta_1 = 0.95$, | $\mathrm{lr} = 5\mathrm{e}{-3}$ |
| | SGD | **65.94** | $\beta_1 = 0.95$, | $\mathrm{lr} = 5\mathrm{e}{-1}$ |

In addition to Figure 3, we report the training perplexity for all other batch sizes in Figure 3. We repeat the experiments from Section 2.3 to verify that our findings generalize to a different dataset and a deeper model.

### B.1  CLIPPING ACTS DIFFERENTLY AT DIFFERENT BATCH SIZES

By detailed analysis of runs from Figure 1, we observe that gradients are clipped more frequently when training with SGD at large batch sizes, as shown in Figure 11. Additionally, at small batch sizes, SGD performs equally well even without clipping; instead, at large batch sizes, training diverges if clipping is not employed.

### B.2  SCALING EXPERIMENTS ACROSS MODEL SIZES AND DATASETS

We train the same 12-layer Transformer on the Fineweb dataset using SGD with momentum and Adam, tuning the learning rate as explained in Appendix A. Batch sizes vary from 4 to 512, and we use 3 different run lengths (i.e., different token budgets). From Figure 15, we observe that, at a fixed number of steps, the performance gap increases with batch size, and that with smaller batches and sufficiently long training, SGD outperforms Adam, consistent with the findings reported earlier.

In a second experiment, we increase the model depth to 24 layers while keeping all other settings identical to Section 2.3. We vary batch sizes from 4 to 64 and training lengths, and tune the learning rate as explained in Appendix A. As shown in Figure Figure 16, the same pattern holds for a deeper model.

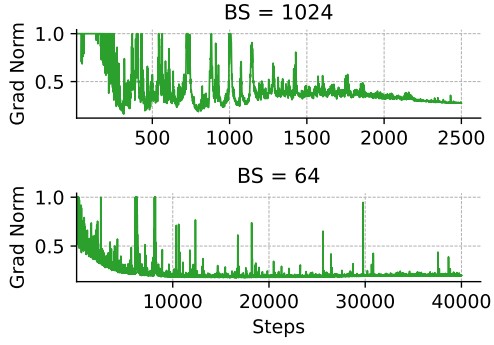

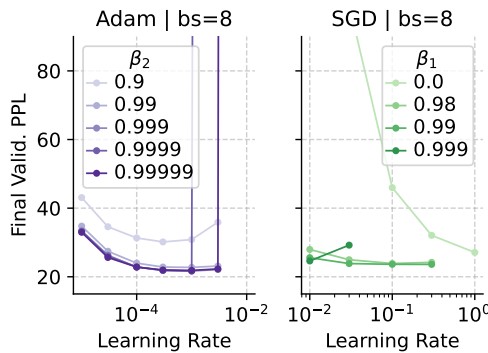

Figure 11: Gradient norm after clipping (threshold 1.0) shows that clipping is more frequent in large-batch training. The setup for these runs is the same as Figure 1.

Figure 12: Final perplexities for batch size 8, shown in addition to Figure 5 (main text).

### B.3 TUNING ADAM IN SMALL-BATCH SETTINGS

In addition to the results from Section 2.4, we show batch size 8 and full training trajectories. Figure 12 shows the final perplexity for batch size 8 across different $\beta_2$ values. Figure 13 presents the full training curves for batch sizes 1, 8, and 256, illustrating convergence dynamics and the relative performance of Adam and SGD.

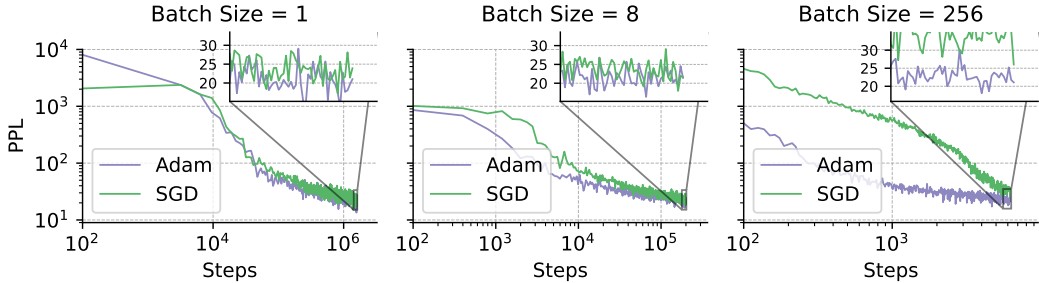

Figure 13: Training dynamics for batch sizes 1, 8, and 256, corresponding to the final perplexities shown in Figure 5 (main text) and Figure 12 (Appendix).

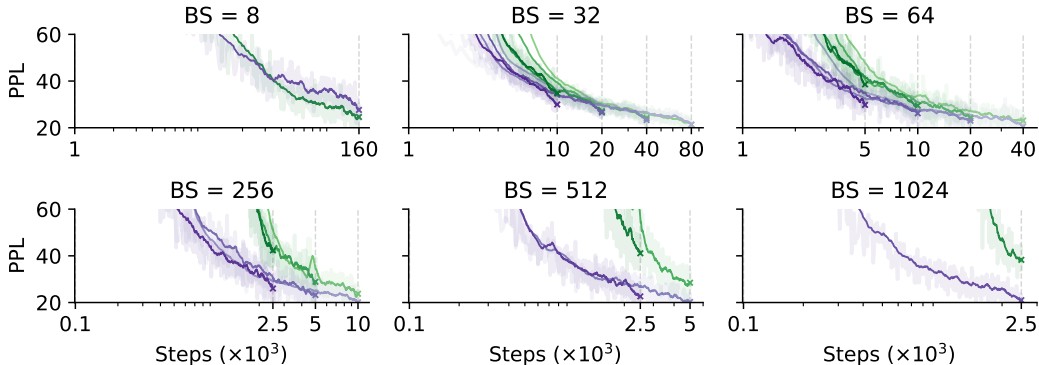

Figure 14: Perplexity during training for SGD (green) and Adam (purple) across different training lengths for all other batch sizes not shown in Figure 3. Solid lines show the rolling mean of PPL values; lighter lines show the raw values. As before, the gap decreases the longer we train, and SGD can eventually outperform Adam.

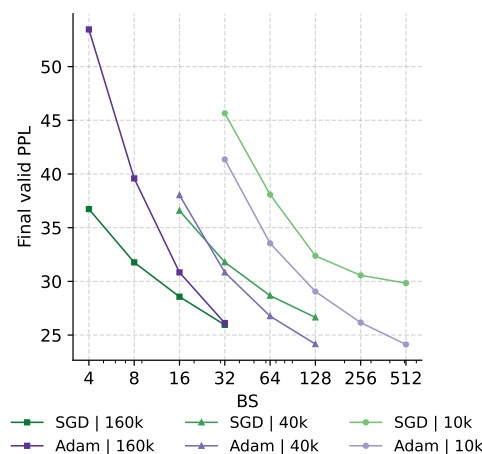

Figure 15: Fineweb dataset, sequence length 2048, 12-layer Transformer.

Figure 16: SlimPajama dataset, sequence length 2048, 24-layer Transformer.

### B.4 ADDITIONAL RESULTS FOR SECTION 4.1

We provide additional experiments supporting Figure 9. In Figure 17, we present results for a least-squares regression problem on the California Housing dataset from scikit-learn across different batch sizes and learning rates. Consistent with Figure 9, batch size affects the progress of signSGD, while SGD performs similarly across batch sizes.

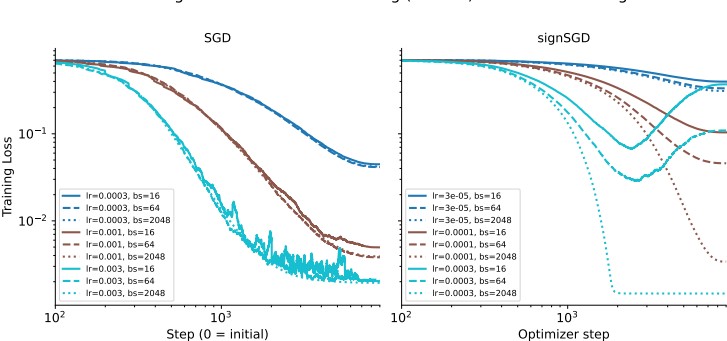

Figure 17: SGD and SignSGD on least-squares regression problem on the California Housing dataset from scikit-learn across different batch sizes and learning rates, supporting Figure 9. Different colors correspond to different learning rates.

## C REVISITING PRIOR EXPLANATIONS: DIRECTIONAL SHARPNESS

Pan & Li (2023) introduce directional sharpness to explain the optimizer gap by studying a second-order Taylor expansion of the loss along the update direction. In this view, the first-order term (gradient correlation) measures how well the update aligns with the negative gradient, while the second-order term (directional sharpness) measures curvature along that direction. Making optimization progress requires a strong negative gradient correlation and low directional sharpness. Let $f$ be a generic loss to optimize and $x_k$ denote the model parameters at iteration $k$, then

$$f(x_{k+1}) = f(x_k) + \underbrace{\nabla f(x_k)^\top (x_{k+1} - x_k)}_{\text{gradient correlation}} + \frac{1}{2} \underbrace{(x_{k+1} - x_k)^\top \nabla^2 f(x_k)(x_{k+1} - x_k)}_{\text{directional sharpness}} + O(\eta^3) \quad (3)$$

In Figure 18, we visualize gradient correlation, directional sharpness, and their sum — a second-order approximation of loss change, to indicate progress. As in our previous analysis, we compare two

settings with drastic performance differences: batch sizes 64 and 1024 from Section 2.2. In the large-batch setting, SGD shows low gradient correlation and high directional sharpness, resulting in weak or even positive total loss change, as reflected in the sum. In contrast, Adam has higher gradient alignment and lower directional sharpness throughout training. When SGD succeeds, its gradient correlation and directional sharpness closely match Adam's, producing a negative loss change in the sum. While these metrics align with SGD's success or failure, they do not directly explain why Adam outperforms SGD, nor why SGD performs well in small-batch regimes.

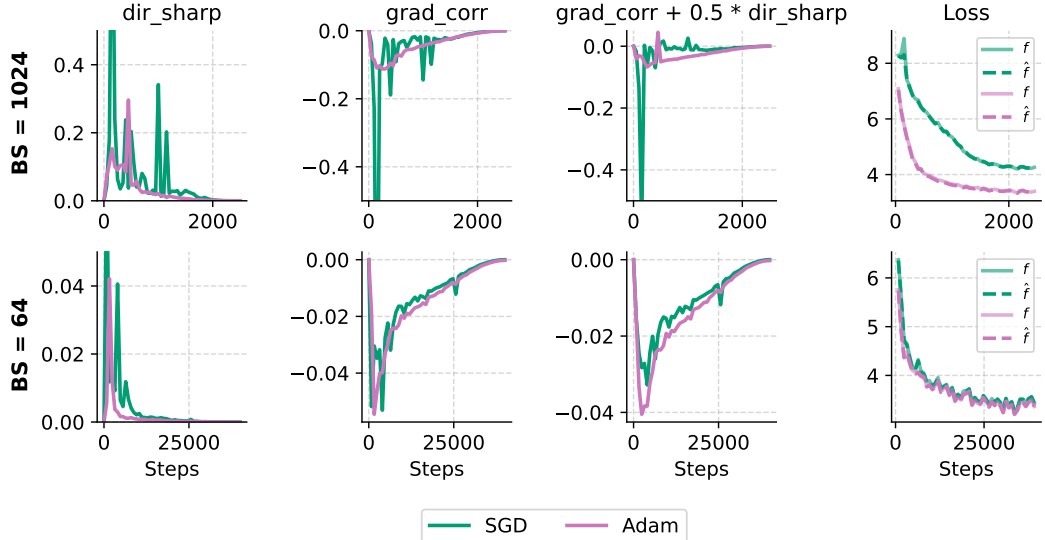

Figure 18: Gradient correlation, directional sharpness, their sum and second-order loss approximation during training, under small- and large-batch settings. Irrespective of the optimizer, good training trajectories have strong negative gradient correlation and low directional sharpness.

As discussed in Section 4, we aim to isolate which component of the SGD update is more problematic in large-batch training. To do that, we focus on the setup from Section 2.2 with batch size 1024 and the optimal $\beta_1$ and learning rate, and perform analysis using grafting and adaptive clipping. The following subsections provide detailed results for each approach.

## D  Insights from Grafting and Adaptive Clipping

### D.1  Insights from Grafting

To isolate the role of update direction and magnitude, we use the grafting technique proposed by Agarwal et al. (2020), which applies the update direction of one optimizer with the magnitude of another. We train the model in the large batch setting, using both combinations: 1) SGD magnitude with Adam direction (SGD#Adam), and 2) Adam magnitude with SGD direction (Adam#SGD). We use the optimal $\beta_1$ from Section 2.2, and sweep the learning rate for the grafted update. We report the training perplexity using the optimal learning rate for both grafting combinations in Figure 19. As shown, using SGD magnitude with Adam's direction performs comparably to Adam, while the reverse combination behaves similarly to SGD. This suggests that the *update direction* is the more problematic component of the SGD update in large-batch training.

### D.2  Insights from Adaptive Clipping

The perspective that direction is the core problem aligns with the observation that global norm clipping does not help much in large-batch training with SGD. If the direction is the main issue, simply rescaling the gradient norm does not lead to better updates.

To investigate this further, we experiment with adaptive clipping, motivated by Pan & Li (2023). As shown in Algorithm 1, we clip the top $p\%$ of the largest momentum coordinates at each step.

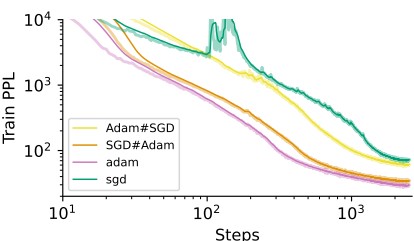 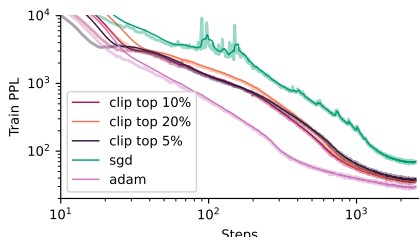

Figure 19: Grafting in large-batch training: using Adam's direction results in performance closer to Adam, while SGD direction leads to results closer to SGD.

Figure 20: Adaptive clipping with different percentages of clipped coordinates in large-batch training. It improves SGD but still does not fully match Adam.

We test several values of $p$ (5, 10, and 20 %). For each value, we keep the optimal $\beta_1$ from the previous setting and tune the learning rate. Clipping with $p = 10\%$ performs best, but we observe that performance does not vary much across different values of $p$. This method helps reduce the gap between SGD and Adam, as shown in Figure 20. This suggests that a subset of larger update coordinates consistently contributes to poor update directions and slows down SGD in large-batch training. In contrast, small-batch training does not present the same problematic coordinates.

Further, we ask whether certain groups of parameters are more likely to produce problematic coordinates. To explore this, we inspect which layers the clipped momentum coordinates come from, using the best-performing setting with $p = 10\%$. In Figure 21, we show the fraction of parameters within each layer that are clipped, relative to the total number of parameters in that layer. We find that normalization layers are clipped the most, which aligns with findings from Zhao et al. (2024) and Tomihari & Sato (2025). However, this does not imply that only normalization layers are problematic. As we observe significant clipping across other layers as well, this suggests that large coordinates persist across all parameters, though they are most pronounced in normalization layers.

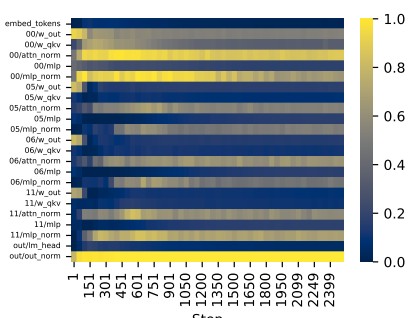

Figure 21: Fraction of clipped momentum coordinates per layer during training, using $p = 10\%$ adaptive clipping. Only a subset of blocks is shown for clarity, as similar patterns are observed across all blocks. Clipping is present across all parameters, but most pronounced in normalization layers.

## E  PROOF SKETCH FOR SECTION 4.1

To build intuition for the proof in (Compagnoni et al., 2025b), we begin by studying the quantity

$$\text{sign}(m(x)), \qquad m(x) \sim \mathcal{N}(\nabla f(x), \sigma^2/B).$$

That is, $m(x)$ is an estimate of the gradient, which we assume for simplicity to have Gaussian distribution centered around the full-batch gradient $\nabla f(x)$. In expectation, we get (coordinatewise)

$$\mathbb{E}[\text{sign}(m(x))]$$
$$= \mathbb{P}[\text{sign}(m(x)) = \text{sign}(\nabla f(x))] \cdot \text{sign}(\nabla f(x)) - \mathbb{P}[\text{sign}(m(x)) \neq \text{sign}(\nabla f(x))] \cdot \text{sign}(\nabla f(x))$$
$$= (2\mathbb{P}[\text{sign}(m(x)) = \text{sign}(\nabla f(x))] - 1) \cdot \text{sign}(\nabla f(x))$$

At this point, the $\text{erf}$ function comes in. Recall that, if $Z \sim \mathcal{N}(\mu, \varsigma^2)$, then if $\ell > \mu$, we have $\mathbb{P}[Z \leq \ell] = \frac{1}{2} + \frac{1}{2}\text{erf}\left(\frac{\ell - \mu}{\sqrt{2\varsigma^2}}\right)$. Note that this implies, for $\mu \geq 0$, $\mathbb{P}[Z \geq 0] = \mathbb{P}[\text{sign}(Z) = \text{sign}(\mu)] = \frac{1}{2} + \frac{1}{2}\text{erf}\left(\frac{\mu}{\sqrt{2\varsigma^2}}\right)$. Hence, if $\nabla f(x) > 0$ for a specific coordinate, $\mathbb{P}[\text{sign}(m(x)) =$

$\text{sign}(\nabla f(x))] = \frac{1}{2} + \frac{1}{2}\,\text{erf}\left(\sqrt{B}\,\frac{\mu}{\sqrt{2\sigma^2}}\right)$, relative to that coordinate. By symmetry of this argument for negative $\nabla f(x)$, we get exactly the drift term in Theorem 1, discussed above:

$$\mathbb{E}[\text{sign}(m(x))] = \text{erf}\left(\sqrt{\frac{B}{2}}(\sigma^2)^{-\frac{1}{2}}\nabla f(x)\right).$$

Finally, note that Gaussianity is not strictly needed for our insights on batch-size acceleration to hold. As discussed by Compagnoni et al. (2025b) and clear from the argument above on the cumulative distribution, a similar expression can hold even for distributions with heavier tails, such as the $t$-student (see Corollary C.10 in Compagnoni et al. (2025b)).

## F TOY QUADRATIC EXAMPLE

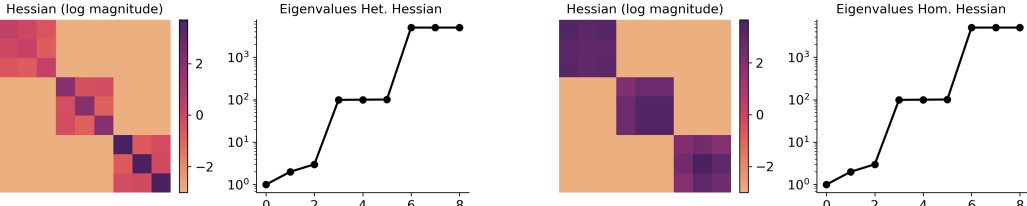

Figure 22: *(left) Heterogeneous and (right) Homogeneous Hessian considered in Figure 8.*

Our setup is inspired from the results and discussions in Zhang et al. (2024a), and uses the codebase of Orvieto & Gower (2025). We consider the loss

$$L(w) = \frac{1}{2}w^\top H w$$

where we construct the Homogeneous and Heterogeneous Hessians using the following procedure:

- We fix the eigenvalues, equal in both cases, to

$$\text{eig}(H_{\text{hom}}) = \text{eig}(H_{\text{het}}) = \{1, 2, 3, 99, 100, 101, 4998, 4999, 5000\}.$$

- We choose both Hessians to be block-diagonal, with blocks of size $3 \times 3$. The homogeneous Hessian has eigenvalues of different magnitude in each block, while the Heterogeneous keeps similar magnitudes in each block.

```
H_details_het = [[1,2,3],[99,100,101],[4998,4999,5000]]
H_details_hom = [[1,99,4998],[2,100,4999],[3,101,5000]]
```

- For each block, we apply a random rotation to the diagonal matrix of eigenvalues, specific to each block. Each rotation is sampled from the Haar measure by decomposition of a random $3 \times 3$ positive semidefinite matrix $AA^\top$, where $A \in \mathbb{R}^{3\times3}$ has i.i.d. Gaussian entries.

The result is shown in Figure 22. Leraning rates for each method are tuned.

Next, to introduce stochasticity in this setting, we simply take the square root of the Hessian to define a $9 \times 9$ design matrix $X$:

$$H = X^\top X, \qquad X = H^{\frac{1}{2}},$$

and subsample a number (the batchsize) of rows of $X$ at each iteration.

Additional learning rates for Figure 22 are reported in Figure 23.

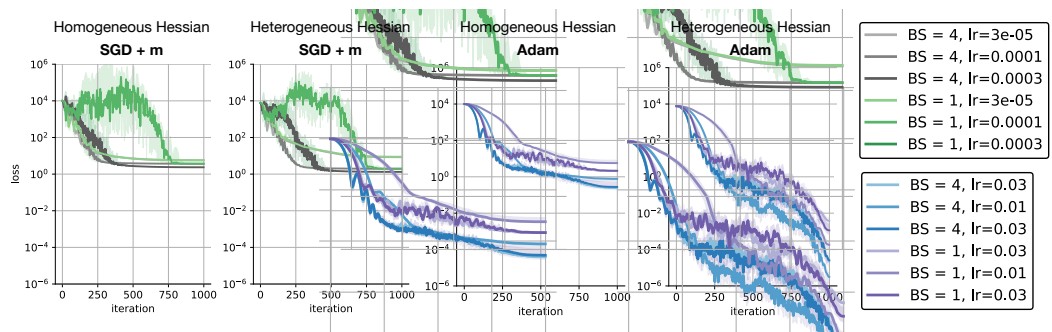

Figure 23: Complement to Figure 8.

# G ALGORITHMIC DETAILS

---

**Algorithm 1** SGD with Adaptive Momentum Clipping

---

**Require:** Initial point $x_0$, learning rate $\eta$, momentum $\beta$, clipping fraction $p \in (0, 1)$
1: **for** $t = 1$ to $T$ **do**
2:      $g_t \leftarrow \nabla f(x_t)$
3:      $m_t \leftarrow \beta m_{t-1} + g_t$
4:      Set clipping threshold $\tau_t$ as the $(1 - p)$-quantile of $|m_t|$
5:      $\hat{m}_t \leftarrow \text{clip}(m_t) = \text{sign}(m_t) \cdot \min(|m_t|, \tau_t)$
6:      $x_{t+1} \leftarrow x_t - \eta \hat{m}_t$
7: **end for**

---

