# OpenReview forum: "Is your batch size the problem? Revisiting the Adam-SGD gap in language modeling"
_ICLR.cc/2026/Conference — Submitted to ICLR 2026_

### Official Review · Reviewer_Hh7C · 2025-10-17

**Soundness:** 2
**Presentation:** 2
**Contribution:** 2
**Rating:** 2
**Confidence:** 4

**Summary:**

This paper studies the setting where SGD can match the performance of Adam by conducting extensive hyperparameter tuning. This perspective helps verify whether previous explanations of Adam’s advantage over SGD are effective regardless of choice of batch size. They also use the tool of SDE to show how batch size can affect SGD and signgd in different ways.

**Strengths:**

1. The author tried small batch experiments in various setting to test whether previous theoretical explanations hold. Most experiments are conducted in a scientific and fair way with ablation study.
2. The result that SGD won’t break and can even match the performance of Adam with much more iterations is a bit novel. I am not aware other papers that actually run SGD for so many iterations.

**Weaknesses:**

1. The main finding lacks novelty. Kunster et al.2023 already shows in their figure 3 and figure 4 that the gap between Adam and SGD decreases or even disappears at small size while this paper also notes the consistency and similarity with previous results.
2. The setting of small batch size and more update iterations is practically irrelevant and this paper also admits that this setting is inconvenient in training LLMs. Therefore, this paper doesn’t help to answer this important question that why Adam outperforms SGD in the conventional LLM training setting. And it is a bit unfair to directly compare the large batch size and small batch size under the same compute budget because the small batch size setting can take many more iteration steps.
3. Though the experiment results clearly shows SGD can match Adam’s performance in small batch size setting, it can’t strongly support the author’s other claims. For example, I don’t think it falsifies the Hessian heterogeneity explanation. Please see the detailed question below. And Kunster et al. 2024 also mentions that they do not attempt to quantify the interaction between stochasticity and class imbalance. So I think it is unfair to claim all the previous explanations struggle to explain the phenomenon in this paper. And they still make sense in the setting of large batch size setting, which is arguably the more practical setting than small batch size.
4. The theory and claims in section 4.1 don’t make sense to me. First they only cite theorems and proof from other paper without proving anything new, so it can’t be viewed as a significant contribution of this paper. Second, I think the way interpreting the result is problematic. See questions below.
5. The gradient clipping and learning rate grafting is disconnected to the major claims. They provide the intuition that gradient scale isn't the problem of SGD in the large batch size setting, which can be a good start of another paper. But I don't see how it contributes to this paper.

**Questions:**

1. As mentioned in weakness 3, I don’t understand why the authors interpret Zhang’s result as explanation invariant to batch size. The effect of hessian definitely gets boosted when the noise is small. When there is more noise, the effect of hessian structure becomes less significant. So I think the experiment results in this paper can still be explained under the Hessian heterogeneity framework.
2. As mentioned in weakness 4, I don’t understand the interpretation of theorem 1 and figure 9. Why does a larger drift term indicate faster convergence? A faster convergence rate needs to be rigorously proved by showing the dependence on T and noise level(batch size). Also I don’t understand why the authors conclude from figure 9 that SGD in early training is dominated by the drift term. Why do you use the same learning rate and how do you choose 1e-3? Also I think it is natural that signgd won’t perform well in the small batch size setting because the sign function can provide completely opposite/wrong update direction when the noise is too large. So I don’t understand how figure 9 can support the claim from SDE.
3. In appendix C, how do you compute directional sharpness when it involves a huge Hessian matrix? Also for figure 17, I feel the difference between Adam and SGD in the small batch size setting is not as significant as you claimed. For Adam, the sum of two terms is also close 0 or sometimes becomes positive. So I don’t think the metrics are even aligned with the optimizer’s success or failure.

---

> ### Author Response · Authors · 2025-11-21
>
> We thank the reviewer for highlighting both the quality of our experiments and the novelty of showing that SGD can match the performance of Adam at small batch sizes. Since novelty is also raised as a concern, we first clarify how our findings differ from prior work.
>
> ---
> ### Comparison with Kunstner et al. (2023)
>
> Kunstner et al. (2023) (https://arxiv.org/pdf/2304.13960 ) also studied the impact of batch size on the Adam–SGD performance gap. However, their focus is primarily on the full-batch setting, showing that noise is not the key contributor and that the root of the gap is deterministic, with transformers (not just language models) exhibiting a performance gap even without noise. Although they examined the effect of batch size in regimes similar to ours (Figures 3 and 4 in their paper), they did not identify or analyze the regime where both Adam and SGD perform well, and the small-batch regime that is the focus of our work is not explored. Their main focus is to understand how the gap evolves as batch size increases - observing that it grows with batch size in language models and drawing connections to SignSGD. There is no discussion on whether the gap shrinks as batch size decreases, and this effect is not clearly visible from Figure 4 in their paper.
>
> Additionally, all their experiments are conducted at a small scale. The only LLM setup in Kunstner et al. is character-level language modeling with sequence length 128 on Wikitext-2, for 40–320 epochs. While their findings are interesting and a **strong motivation for this paper**, they cannot serve as conclusive evidence to invalidate our investigation.
>
> Furthermore, Figures 3 and 4 in their paper show that the gap shrinks across different batch sizes but do not show that SGD trained for sufficient iterations can match a large-batch Adam run. In their results, the gap persists, black curves never reach the level of red curves, whereas our Figure 4 shows that the order can actually flip.
>
> On a more theoretical note, character-level loss does not capture the gradient noise structure and potential heavy-tail behavior believed to be important for understanding the Adam–SGD gap. **Our findings complement those of Kunstner and offer evidence and theoretical insights on the shrinkage of the Adam-SGD gap as the batch size increases.**
>
> ---
> ### Explaining the gap
> As stated in the introduction, this paper's purpose is not to explain the Adam-SGD gap but to eliminate confounders. We believe – and indeed showcased in Figure 8 – that **Hessian Heterogeneity is a main driver of the Adam - SGD gap, especially in the large batch size setting.** This, however, **does not alone explain** why the batch size is a primary driver of the Adam-SGD gap. While we agree (and state!) that small batch size training is not useful in large-scale setups, we believe our paper refines the Hessian heterogeneity explanation, as well as other explanations, offering concrete evidence at a scale which is uncommon for optimization papers.
>
> Stated in other words, what we debunk is the simple causal arrow: *heterogeneity -> gap* or *class imbalance -> gap*. This is not correct and holds only at large enough batch sizes. Crucially, this is not because small batch sizes are “bad for all methods”, but because (as we explain in the last section) the full preconditioning power is only unlocked at full batches. While optimization experts might foresee that the advantages of preconditioning are only evident at large batch sizes (as in the classic literature on stochastic Newton), we believe it was not clear before our investigation that SGD (with momentum) could also achieve similar performance.
>
> ---

---

> > ### Author Response · Authors · 2025-11-21
> >
> > ### SDE approach
> > We do not believe that using a theoretical result from previous work is a limitation. We do not claim novelty (besides a simpler intuitive proof in App E). Regarding our interpretation of this result, let us be more thorough and explain this with more precision. In Figure 9, the choice of learning rate is arbitrary: we **do not claim** that SignSGD performs better or worse than SGD here (it depends on the learning rate, as you say!). What we want to draw attention to is the following: **for a fixed learning rate, varying the noise level leads to nearly identical loss curves for SGD (classical curvature-dominated regime)**. In contrast, **for signSGD the effect of batch size is visible from the very first iterations**. This distinction (which again does not imply that SignSGD is faster than SGD!) is due to the drift term: in early iterations, as the SGD runs confirm, the effect of noise is minimal, and what one truly sees is the impact of a batch-dependent drift term.
> >
> > The reviewer is, however, right in claiming that this is not proving that SignSGD suffers from small batch – it is indeed just an indication that batchsize affects dynamics in early training. This is a crucial point, but not the only one the analysis can offer. In Appendix E, we show (compressing Compagnoni’s proof) that the expectation of a noisy gradient model $sign(g)$, where $g$ is Gaussian distributed around $\nabla f(x)$ with variance $\sigma^2/B$, is $\operatorname{erf}\left(\sqrt{\frac{B}{2}}(\sigma^2)^{-\frac{1}{2}}\nabla f(x)\right)$. Note that as $\sqrt{\frac{B}{2}}(\sigma^2)^{-\frac{1}{2}}$ approaches $0$ (i.e., shrinking the batch size), then $\operatorname{erf}\left(\sqrt{\frac{B}{2}}(\sigma^2)^{-\frac{1}{2}}\nabla f(x)\right)\approx \sqrt{\frac{B}{2}}(\sigma^2)^{-\frac{1}{2}}\nabla f(x)$. This indicates that not only is early-stage training batch-size-dependent, but also that small batches imply a latent model that is non-adaptive. We hope this can clarify the reviewer’s doubts about this section.
> >
> > ---
> > ### Directional sharpness
> > To calculate directional sharpness, we use the Hessian-vector product from the PyHessian codebase (https://github.com/amirgholami/PyHessian/blob/de8cf239ace21c7754ee599255897ebf3eed19cb/pyhessian/hessian.py#L63). Regarding Figure 17 (18 in revisited version), for good training trajectories we expect strong negative gradient correlation and low directional sharpness. The difference in these metrics between Adam and SGD in the **small-batch** setting is negligible (bottom row). On the other hand, in the large-batch setting (upper row), the difference is more persistent, showing "better" metrics for Adam – not at every iteration, but for the most of training. Thus, as noted in the paper, these metrics align with optimizer success but do not directly explain the gap..
> >
> >
> > ---
> > ### Clipping and LR grafting
> > The ablations on gradient clipping and learning rate are done to understand what goes wrong for SGD in large-batch settings that doesn't happen with small batches – i.e. what problems small batches seem to solve. Clipping and LR grafting experiments together show that **small-batch SGD corrects problematic update directions** seen in large-batch training. These results therefore are not disconnected from the main claims: they complement the main findings by isolating the cause of large‐batch failure and also suggesting directions for future work.
> >
> > ---
> >
> > We hope this response clarifies the novelty of our findings, makes the connection to prior work and the limitations of their explanations clearer, and better explains our contribution to understanding the Adam-SGD gap. We would appreciate it if the reviewer could revisit the concerns raised in the weaknesses and continue the discussion.

---

### Official Review · Reviewer_Kxeu · 2025-10-25

**Soundness:** 3
**Presentation:** 3
**Contribution:** 3
**Rating:** 4
**Confidence:** 4

**Summary:**

This pager studies the gap between SGD and Adam as the batch size varies. For language model training, it finds that the gap can be small, but only when the batch size is very small. Various existing theoretical models do not explain this phenomenon well, but the paper suggests a simple comparison of how noise impacts SDE approximations of SGD vs. SignSGD provides some explanation.

**Strengths:**

1. The paper presents an interesting observation about training dynamics in the small batch regime that is not easily explained by some existing models. In particular any model that suggests Adam is better than SGD needs to account for why that property does not hold at small batch sizes.
2. The experiments seem to be rigorous with appropriate tuning and ablations.

**Weaknesses:**

1. The SDE explanation is not really fully fleshed out. First, in the figure, it is the noise level not the batch size that is varied (I understand that they are connected, but it should be possible to make an experiment with batch if so). Moreover, it is not clear that the gradient noise model is a good one in the full language modeling case. If this is the main explanation provided by the paper, there should be some more clear experiments trying to substantiate the model on real data. For example, looking at gradient or update variance could be important. Last, the analysis does not account for momentum (which I know is challenging), but seems important since the SGD results in particular require very high momentum.
2. All the experiments are pretty under-trained. Only doing 1.3B tokens on a 160M model is even below chinchilla (20x), and way below the practical token-to-parameter ratios encountered in practice. This seems important since most optimization will happen later in training, so only studying the early training regime may give misleading results. I would suggest running some of the main results longer to see what happens.

**Questions:**

1. Why are experiments done with no weight decay?
2. In Figure 9, is the batch size fixed across runs?

---

> ### Author Response · Authors · 2025-11-21
>
> We thank the reviewer for noting that our observation is interesting and our ablations were performed correctly. We believe all concerns can be quickly resolved, and would be glad if the reviewer could revise their evaluation.
>
> ---
> ### The noise level and batch size
>
> Regarding Figure 9: as the reviewer already noted, gradient noise is inversely proportional to the batch size, as classical analysis based on sampling reveals (e.g. formula 4 here https://arxiv.org/pdf/1711.04623). **Varying the noise level is therefore perfectly equivalent to varying the batch size**. To make our point more straightforward, we included a least-squares regression example with real data (sklearn, California Housing) in Appendix B.4 of our revised version. Note that we also disagree with the claim that this setting fully models LLMs: indeed, SignSGD is much slower here! Our point, however, is another and more fundamental: **SGD has early-training performance that is batch-size independent - in contrast to SignSGD, where early-training speed is affected by batch size**.
>
> ---
> ### Token budget
>
> It is true that in Section 2.2 we use a token budget of 1.3B, but it is incorrect that all experiments are under-trained. **We have setups up to 2x the Chinchilla-optimal** token budgets in the paper. In fact, our largest ablation trains up to 5.5B tokens (Figure 2). There we show that the gap shrinks with more iterations at a fixed batch size. Thus, the conclusions from Figure 1 would remain the same if trained longer, and the gap would simply be smaller across all batch sizes.
>
> ---
> ### Regarding the questions
>
> 1. We do not operate with weight decay to eliminate this confounder (and the need for further tuning) from our setup. We do not expect or have reasons to believe weight decay to be a main driver of the Adam-SGD gap
>
> 2. The noise in Figure 9 is injected and is at different scales. This corresponds to having different batch sizes. Please see our new Figure 17 in Appendix B.4 for a real-dataset setup where we inject no noise but consider different batch sizes instead. Findings are similar.

---

### Official Review · Reviewer_yBqU · 2025-10-31

**Soundness:** 1
**Presentation:** 1
**Contribution:** 1
**Rating:** 2
**Confidence:** 4

**Summary:**

The paper revisits the question of the reason behind the gap in the performance of Adam and SGD, by studying examples where SGD performs comparably to Adam. It finds that the gap decreases at small batch sizes. It also finds that various factors such as Hessian heterogeneity and class imbalance can exist at small batch sizes, but the gap is still low. Finally, it proposes a theory based on SDE approximation of SGD and SignSGD for explaining the gap.

**Strengths:**

The paper tries to understand the gap between Adam and SGD based on the observation that the gap diminishes at small batch sizes.

**Weaknesses:**

The observation that Adam and SGD gap should diminish at small batch sizes is already supported by various previous theoretical works [1,2]. These papers show that the benefits of preconditioning decrease at small batch sizes, even for quadratics. Thus, the observation is expected.

Secondly, first half of the paper provides misleading results as the $\beta_2$ of Adam is not properly tuned for small batch sizes. In my opinion, this is a significant misrepresentation for the first half of the paper, and should be either removed or clearly stated beforehand that the gap in the following section vanishes after tuning $\beta_2$.

[1] - Zhang et al. 2019 - Which Algorithmic Choices Matter at Which Batch Sizes? Insights From a Noisy Quadratic Model

[2] - Jain et al. 2018 - Accelerating Stochastic Gradient Descent For Least Squares Regression

**Questions:**

The definition of critical batch size as mentioned in lines 455-464 seems non-standard. The critical batch size is defined as the maximum batch size till which we can linearly decrease the number of steps required to achieve a given loss. Are the two definitions the same? To clarify, for SGD, critical batch size is not 1, and the SDE can be preserved while scaling up batch size by scaling eta proportionally to batch size. Is the claim that the batch size for signSGD will precisely be the B for which the erf linear scaling fails? Can this be proven?

---

> ### Author Response · Authors · 2025-11-21
>
> We thank the reviewer for the opportunity to clarify some of our claims.
>
> ---
> ### Critical batch size
>
> Most of the confusion stems rightfully from the large body of literature on critical batch sizes, perfect scaling, and speed-noise (train/test) trade-off in classical neural network setups trained for several epochs. LLM pretraining is instead (in most academic cases) about < 1 epoch performance, where train and test losses are similar and no classical trade-off driven by convergence to the stationary distribution (as in the setting of Zhang et al.). For this setup, much less is known about critical/optimal batch sizes, and the topic is the object of recent research (see e.g. the paper “How Does Critical Batch Size Scale in Pre-training?” (https://arxiv.org/abs/2410.21676) and “Scaling Law for Language Models Training Considering Batch Size” (https://arxiv.org/abs/2412.01505 ).
>
> In Section 2.3, we discuss critical batch size empirically in the sense that, under a fixed token budget, increasing batch size for Adam (i.e. reducing the number of update steps) preserves ppl up to a certain point. In contrast, for SGD we observe performance degradation as batch size increases, starting from very small batch sizes, suggesting CBS for SGD is close to 1 here. Following this, in Section 4.1 we draw a connection to the critical batch size for signSGD: increasing batch size B improves the drift, up to the saturation point of the erf drifts which corresponds to CBS.
>
> ---
> ### Novelty in comparison to Zhang et al. (2019)
>
> We took the time here to thoroughly and precisely explain **why our setup, insights, and findings differ from those of Zhang et al**. Since this is the reviewer’s central point for rejection, we would be glad if their discussion could be revised.
>
> Let us start reviewing Zhang et al. (2019) : they explore the trade-off between the speed of convergence to the steady-state risk and the value of the steady-state risk in quadratic problems optimized with SGD or preconditioned methods using deterministic preconditioners (important! We will go back to this later, in point 3 below). For SGD, the loss evolves (using the notation of Zhang et al.) as
>
> $$O\left( (1-\alpha h_d)^{2t} \right)+O\left( \frac{\alpha}{B} \right),$$
>
> While for deterministic Hessian-based preconditioning (stronger for higher values of $p$), it is
>
> $$O\left( (1-\alpha h_d^{\,1-p})^{2t} \right) + O\left( \frac{\alpha}{B}\, h_d^{-p} \right).$$
>
> Quoting directly Zhang et al., “traditional acceleration techniques (e.g., momentum and preconditioning) help improve the convergence rate at the expense of increasing the steady state risk. Therefore, the NQM implies that momentum and preconditioning would benefit more from large-batch training compared to plain SGD.”
>
> While on the surface and without the proper context, this claim depicts what is happening in our experiments, our mathematical setup and the learning problems **are not similar** to Zhang at al. Evidence is also in the literature progression: since 2019 (the year of Zhang et al.), a large number of papers have been written on the Adam-SGD gap, specifically in transformers: this setup is clearly different from the pre-LLM ones and is purely understood. Our theoretical setup in the last section is indeed drastically different from Zhang et al., and conclusions are driven by another fact: the speed of convergence (not error floor!) is affected by batch size. Zhang et al. did not study this effect, since their theory is limited to deterministic preconditioners.

---

> ### Author Response · Authors · 2025-11-21
>
> Let us go step by step.
>
> 1. In LLMs, we are not in the “tradeoff setting” between speed and noise that is studied by Zhang et al. and is observed in neural networks trained for hundreds of epochs. We are in an online setup where we see 1% of the dataset (SlimPajama is 627B tokens), and only once. We could indeed train such models for many more steps without saturation. The theory of Zhang et al. instead covers the trade-off setup, where noise can dominate since training runs for several epochs. Their theory and discussion concern only this tradeoff, in which the error floor is the only quantity affected by the batch size. In equation 2 and Appendix E, we show that in stochastic sign-based preconditioning, the speed is also affected by the batch size.
>
> 2. It is well known (since indeed we train for <1epoch) that train and test losses in language modeling are similar at standard compute scales. Our results (e.g., see Figure 3) concern both train and test performance. Zhang et al. mainly consider test performance, and inspect train performance in their Figure 8: for a target accuracy of 0.99 and 0.83, **they do not observe any gap in the adam-sgd training performance**. This is to expect, since the error floor value is known to affect generalization (see e.g.https://arxiv.org/abs/1711.04623)
>
> 3. Our theoretical motivation for this phenomenon is that, as equation 2 depicts, stochastic preconditioning has a convergence speed that is affected by noise (i.e., by batch size value). Figure 9 provides a simulation showcasing this. Mathematically, Zhang et al. only consider in their theory algorithms with associated SDE (cf. theory formula 8)
>
> $$
> dX_t = -P^{-1} \nabla f(X_t)\, dt + \sqrt{\frac{\eta \Sigma}{B}} dW_t.
> $$
>
> Where clearly the convergence speed to the error floor is not affected by the batch size (classical analysis). Instead, the SDE model by Compagnoni et al. (2024) reveals that for signSGD
>
> $$d X_t =- \text{erf} \left(\sqrt{\frac{B}{2}} \Sigma^{-\frac{1}{2}} \nabla f(X_t) \right) dt + \text{term}  d W_t.$$
>
> Hence speed – from first iterations and not when the tradeoff becomes apparent – is affected by batch size. Figure 9 shows exactly this.
>
> ---
> ### Effect of tunning beta2
>
> In most papers concerning Adam and the Adam-SGD gap, betas are not tuned. On the one hand, this is clearly a limitation, on the other it depicts the modern use of the Adam optimizer: see e.g. all models trained in the Pythia repository (https://github.com/EleutherAI/pythia/tree/main/models): they all use betas = 0.9, 0.95. Our choice for our most extensive ablation (Figure 2) is following common strategies, and in particular the suggestion from the critical batch size scaling paper (https://arxiv.org/pdf/2410.21676): beta1=beta2=0.95.
>
> Since we are aware that a larger beta2 can improve performance at low batches, even if not standard, we provide an ablation in Figure 5. Note that while this makes Adam perform better, our point stands the same: **The Adam-SGD gap drastically shrink**

---

### Comment · Area_Chair_TPgu · 2025-11-28

Dear Reviewers,

The discussion phase is now underway, and the authors have finished uploading their responses to reviewers. If you haven't already, please carefully review the authors' responses to understand their perspectives. Engage in thoughtful, constructive discussions with authors, sharing your thoughts and seeking clarifications. Please also update your review or rating if necessary.

It is noted in the guideline that reviewers can leave comments visible to authors **until Dec 2 11:59pm AoE**. Your active participation and contribution to the ongoing discussion are highly encouraged. Thank you very much for your contribution to ICLR.

Best regards,

AC

---

### Meta-Review · Area_Chair_tQek · 2026-01-08

**Summary:**

The reasons behind good performance of Adam for training LLMs has been a question that received significant interest over the last years. The paper aims to obtain a better understanding on this matter by studying how batch size affects the gap between SGD and Adam in modern transformer training. The work finds that SGD can perform similar to Adam in the small batch size setting (if tuned correctly), and finds that prior speculations or analyses do not fully explain this phenomenon. An theoretical analysis on simple quadratic model and using stochastic differential equations aim to bridge the gap.

Initially, reviewers gave unanimously low ratings for this submission, mainly due to perceived lack of novelty over previous works, lack of novelty in the SDE anaylsis ("not fully fleshed out") and various criticisms with the experimental setup, hyperparameters, etc.  I found the author's rebuttal to be well-written and clearly addressing most of these concerns. Nevertheless, I share reviewers opinion about overall lack of novelty of the work. Given that the batch-size gap between SGD and Adam and also (quasi)-Newton methods has been known for a while,  I share reviewer's sentiment that the quadratic model and SDE analysis fall a bit short of bringing in new understandings and making substantial progress beyond a purely empirical revisit in modern LLM settings.

This was unfortunately a difficult borderline decision, but the paper is not recommend acceptance at this stage. Further fleshing out the theoretical analysis or crystallizing out better the truly new findings compared to past works will make this paper a clear accept for a  future submission.

**Reviewer Concerns:**

Reviewers were concerned about:
(1) lack of novelty compared to previous works (yBqU, Hh7C)
(2) shallowness and novelty of the SDE analysis (Kxeu)
(3) issues with the experiments / hyperparameters (Kxeu, yBqU)

Point (3) has been addressed by the rebuttal, and points (1), (2) only partially and remain outstanding to some extend.

**Reviewer Scores:**

All reviewers (yBqU, Hh7C and Kxeu) would have likely increased their score, but not significantly to overall reach across the accept threshold.

---

### Decision · Program_Chairs · 2026-01-26

Reject